# Synthesis of Rottlerone Analogues and Evaluation of Their α-Glucosidase and DPP-4 Dual Inhibitory and Glucose Consumption-Promoting Activity

**DOI:** 10.3390/molecules26041024

**Published:** 2021-02-15

**Authors:** Yinan Zhang, Haibo Wang, Yan Wu, Xue Zhao, Zhihong Yan, Robert H. Dodd, Peng Yu, Kui Lu, Hua Sun

**Affiliations:** 1China International Science and Technology Cooperation Base of Food Nutrition/Safety and Medicinal Chemistry, College of Biotechnology, Tianjin University of Science & Technology, Tianjin 300457, China; 15302138535@mail.tust.edu.cn (Y.Z.); 1411502122@mail.tust.edu.cn (H.W.); wuyan6@163.com (Y.W.); zhaoxue1632014@163.com (X.Z.); yanzhihongzs@163.com (Z.Y.); robert.dodd@cnrs.fr (R.H.D.); yupeng@tust.edu.cn (P.Y.); 2Centre de recherche de Gif-sur-Yvette, Institut de Chimie des Substances Naturelles, UPR 2301, CNRS, Avenue de la Terrasse, 91198 Gif-sur-Yvette, France

**Keywords:** rottlerone, α-glucosidase inhibitor, DPP-4 inhibitor, inhibitory mechanism, glucose consumption

## Abstract

Our previous study found that desmethylxanthohumol (**1**) inhibited α-glucosidase in vitro. Recently, further investigations revealed that dehydrocyclodesmethylxanthohumol (**2**) and its dimer analogue rottlerone (**3**) exhibited more potent α-glucosidase inhibitory activity than **1**. The aim of this study was to synthesize a series of rottlerone analogues and evaluate their α-glucosidase and DPP-4 dual inhibitory activity. The results showed that compounds **4d** and **5d** irreversibly and potently inhibited α-glucosidase (IC_50_ = 0.22 and 0.12 μM) and moderately inhibited DPP-4 (IC_50_ = 23.59 and 26.19 μM), respectively. In addition, compounds **4d** and **5d** significantly promoted glucose consumption, with the activity of **5d** at 0.2 μM being comparable to that of metformin at a concentration of 1 mM.

## 1. Introduction

Diabetes mellitus is a multi-gene metabolic disease that is characterized by hyperglycemia that is caused by a relative (Type 2 Diabetes) or absolute (Type 1 Diabetes) insulin deficiency. The number of diabetes patients has been increasing rapidly in recent years due to changes in people’s living standards and dietary habits. The number of adult diabetes patients worldwide is about 463 million, and this number may reach 578 million by 2030, according to the statistical results of the International Diabetes Federation (IDF) in 2019 [1]. Diabetes is becoming one of the most widespread health burning problems in the elderly [2].

α-Glucosidase plays an important role in the physiological process of postprandial digestion and the absorption of carbohydrates in food in patients with Type 2 Diabetes, and it is the key enzyme involved in the hydrolysis of dietary carbohydrates [3]. α-Glucosidases specifically hydrolyze the α-1,4-glucopyranosidic bond to release glucose. Therefore, the inhibition of digestive α-glucosidase is one therapeutic approach to slowing down carbohydrate digestion and glucose absorption, thereby stabilizing blood glucose level and preventing hyperglycemia in diabetic patients. α-Glucosidase inhibitors have become a better choice in the case of Asian diabetic patients that are associated with high consumption of carbohydrates in their staple diet [4]. Dipeptidyl peptidase-4 (DPP-4) inhibitors enhance the plasma level of active glucagon-like peptide-1 (GLP-1) and glucose-dependent insulinotropic polypeptide (GIP), which results in increased insulin secretion and decreased glucagon secretion. It has been reported that the addition of a DPP-4 inhibitor to patients with type 2 diabetes that is inadequately controlled by an α-glucosidase inhibitor achieved better glycemic control without further increasing the risk of weight gain and hypoglycemia [5]. Thus, the development of dual inhibitors of α-glucosidase and DPP-4 may provide a novel approach to the discovery of new antidiabetic agents. To the best of our knowledge, such α-glucosidase and DPP-4 dual inhibitors have never been reported. It should be noted that the development of new α-glucosidase inhibitors has been largely neglected, since the introduction of acarbose, voglibose, and miglitol leave much room for the improvement of anti-diabetic therapies [3].

Desmethylxanthohumol (**1**, Figure 1), isolated from hops, has been found to display anticancer and antioxidant activity [6,7]. In a previous study, we found that desmethylxanthohumol inhibits α-glucosidase in vitro [8]. Furthermore, our recent investigations revealed that dehydrocyclodesmethylxanthohumol (**2**) and its dimer analogue rottlerone (**3**) [9] exhibited more potent α-glucosidase inhibitory activity than **1** (Figure 1). Therefore, we prepared a series of rottlerone derivatives **4a**–**4d** and **5a**–**5d** for the purpose of studying structure-activity relationships and improving the α-glucosidase inhibitory potencies of compounds **1**–**3**. In addition, the synthesized compounds were also evaluated for DPP-4 inhibition with the goal of identifying potential dual-targeted anti-diabetic drugs.

## 2. Results

### 2.1. Synthesis and Characterisation of Rottlerone Analogues

The synthesis of rottlerone analogues **4a**–**4d**, **5a**–**5d,** and **12**–**13** was accomplished, as shown in Scheme 1. The mono-MOM protected 2’,4’,6’-trihydroxyacetophenone (**6**) was synthesized from 2’,4’,6’-trihydroxyacetophenone according to a known procedure [10]. Compound **6** was converted to **7** by cyclization with 3-methyl-2-butenal, followed by removal of the MOM protection to provide **8**. Conversion of the latter to dimer **9** was achieved by reaction with formaldehyde [11]. The rottlerone derivatives **4a**–**4h** were then obtained by base-catalyzed aldol condensation of **9** with the corresponding substituted benzaldehydes. Deprotection of **4e**–**4h** using 2M hydrochloride acid afforded **5a**–**5d**. Finally, methylated rottlerone derivatives **12** and **13** were obtained by aldol condensation of compound **9** with benzaldehyde followed by *O*-methylation using CH_2_N_2_ in DCM.

### 2.2. Rottlerone Analogues Inhibited α-Glucosidase and DPP-4

The α-glucosidase and DPP-4 inhibitory activities of the synthesized compounds were evaluated. Acarbose and sitagliptin, clinically useful α-glucosidase and DPP-4 inhibitors, respectively, were chosen as the reference compounds for activity comparison. Dimer **9** was first evaluated and compared with the inhibitory activity of compound **2** and rottlerone (**3**), as shown in Table 1. Interestingly, replacement of the styrene moiety of rottlerone by a methyl group (**9**) led to a reduction in α-glucosidase inhibition. The variation of the electronic properties of *para* substituents on the phenyl group did not significantly alter the α-glucosidase inhibitory activity (**4a** and **4b** vs. **3**). However, the introduction of a hydroxy group at this *para* position (**5a**) resulted in a 10-fold improvement in activity. Furthermore, the α-glucosidase inhibitory activities were found to be two-fold and four-fold higher than **5a** in the case of the *o,p*-dihydroxyphenyl and *o,m,p*-trihydroxyphenyl derivatives **5b** and **5d**, respectively. The lower activity of the *o,m*-dihydroxy derivative **5c** is perhaps due to the formation of an intramolecular hydrogen bond between the adjacent hydroxy groups. Notably, compound **5d** also showed a modest DPP-4 inhibitory activity with an IC_50_ = 26.19 μM. Altogether, these results revealed that free phenolic hydroxy groups are crucial for obtaining the desired activities.

Compounds **4c** and **4d**, in which the phenyl ring is replaced by a furan or a thiophene ring, possessed three-fold and 26-fold higher α-glucosidase inhibitory activity than **3**. Noteworthy, compound **3** demonstrated some degree of DPP-4 inhibitory activity (Table 1). This suggests that the phenyl ring of the rottlerone analogues can be advantageously replaced by a heterocyclic ring and it opens the way to a wide variety of further possible structural modifications to improve activity.

The methylation of the phenol hydroxyls on the *2H*-chromene nucleus was also found to have considerable influence on α-glucosidase inhibitory activity. In particular, the trimethylated derivative **13** demonstrated a 10-fold enhancement of this activity when compared to the demethylated analogue **12** and comparable activity to that of **4d** and **5d**. We hypothesize that the 7-methoxy group prevents the formation of an H-bond between the 7- and 7′-hydroxyls, resulting in a vastly different steric environment favorable to α-glucosidase inhibitory activity.

### 2.3. The Enzyme Kinetics of Compounds ***4d*** and ***5d***

The most active α-glucosidase inhibitors of our series, **4d** and **5d**, were then used for kinetic analyses in order to shed light on their mechanism of action. The plots of the velocity versus enzyme concentrations in the presence of different concentrations of **4d** and **5d** gave a family of parallel straight lines (Figure 2a,b). Increasing the inhibitor concentrations did not affect the slopes of the lines, indicating that **4d** and **5d** are irreversible inhibitors. Moreover, the value of 1/V increased with increasing concentrations of **5d**, but the intercept on the X-axis and Y-axis remained constant. This result indicates that compound **5d** is a mixed competitive α-glucosidase inhibitor.

### 2.4. Glucose Consumption of Compounds ***4d*** and ***5d***

In order to further study the anti-diabetic activity of compounds **4d** and **5d**, their effects on cell-based glucose consumption were evaluated using metformin as a positive control. Compound **5d** was observed to significantly promote glucose consumption at 0.2 and 0.04 μM, while compound **4d** showed similar potency at 0.2 μM, as shown in Figure 3.

## 3. Materials and Methods

### 3.1. Chemistry

All of the reagents and solvents were commercially available and used without further purification. The ^1^H-NMR and ^13^C-NMR spectra were recorded on a Bruker AM-400 NMR spectrometer (Billerica, Middlesex, MA, USA) in CDCl_3_ or *d*_6_-DMSO (dimethyl sulfoxide). Mass spectra were obtained on a Q-TOF mass spectrometer (Agilent, Santa Clara, CA, USA) and high resolution mass spectra were obtained on a hybrid IT-TOF mass spectrometer (Shimadzu LCMS-IT-TOF, Kyoto, Japan). Compound purity was determined by UV absorbance at 254 nm during tandem liquid chromatography/mass spectrometry (Agilent, Santa Clara, CA, USA). All of the final products had a purity of ≥95%, as determined using this method.

#### 3.1.1. 1-(2,6-Dihydroxy-4-(methoxymethoxy)phenyl)ethan-1-one (**6**)

1-(2,4,6-Trihydroxyphenyl)ethan-1-one (1.68 g, 10 mmol) in dry CH_2_Cl_2_ (100 mL) were added *N*,*N*-diethylpropan-2-amine (3.93 g, 25 mmol) and chloro(methoxy)methane (1.77 g, 22 mmol) at 0 °C. The reaction mixture was warmed to room temperature and then stirred for 6 h. The reaction was quenched with NH_4_Cl (aq) and extracted with CH_2_Cl_2_ (500 mL) three times. The combined organic layers were washed with brine, dried over Na_2_SO_4_, and concentrated under reduced pressure. The crude product was purified by column chromatography on silica (petroleum ether/ethyl acetate 20:1) to afford 1-(2-hydroxy-4,6-bis(methoxymethoxy)phenyl)ethan-1-one (2.1 g, 82%) as a colorless oil. 1-(2-hydroxy-4,6-bis(methoxymethoxy)phenyl)ethan-1-one (1.28 g, 5 mmol) in MeOH (50 mL) was added 2 N HCl (5 mL) at 0 °C. After addition, the reaction mixture was warmed to 40 °C and then stirred for 3 h. The reaction was quenched with NaHCO_3_ (aq) and extracted with EtOAc (100 mL) three times. The combined organic layers were washed with brine, dried over Na_2_SO_4_, and concentrated under reduced pressure. The crude product was purified by column chromatography on silica (petroleum ether/ethyl acetate 10:1) to afford a compound **6** (0.65 g, 61%) as a white solid. ^1^H-NMR (400 MHz, *d*_6_-DMSO) 12.26 (s, 2H), 6.02 (s, 2H), 5.17 (s, 2H), 3.36 (s, 3H), 2.58 (s, 3H). ^13^C-NMR (100 MHz, *d*_6_-DMSO) 203.8, 164.4, 163.6, 105.8, 95.1, 93.9, 56.3, 33.0.

#### 3.1.2. 1-(7-Hydroxy-5-(methoxymethoxy)-2,2-dimethyl-2*H*-chromen-8-yl) ethan-1-one (**7**)

Compound **6** (2.12 g, 10.0 mmol) in dry pyridine (20 mL) was added 3-methyl-2-butenal (1.26 g, 15.0 mmol). The reaction was heated to 120 °C for 16 h. The mixture was cooled and diluted with CH_2_Cl_2_ (20 mL). The mixture was poured into 5% HCl (aq, 20 mL). The organic phase was washed with NaHCO_3_ (aq) (20 mL) and brine (3 × 20 mL). The organic phase was dried over Na_2_SO_4_ and then concentrated under reduced pressure. The crude product was purified by column chromatography on silica (petroleum ether/ethyl acetate 50:1) to afford a compound **7** (2.09 g, 75%) as a yellow oil. ^1^H-NMR (400 MHz, CDCl_3_) 13.64 (s, 1H), 6.58 (d, *J* = 10.0 Hz, 1H), 6.20 (s, 1H), 5.43 (d, *J* = 10.0 Hz, 1H), 5.20 (s, 2H), 3.47 (s, 3H), 2.67 (s, 3H), 1.50 (s, 6H). ^13^C-NMR (100 MHz, CDCl_3_) 202.4, 164.9, 157.4, 155.4, 123.8, 115.5, 105.5, 102.2, 93.9, 93.2, 77.0, 55.5, 32.2, 26.9.

#### 3.1.3. 1-(5,7-Dihydroxy-2,2-dimethyl-2H-chromen-8-yl)ethan-1-one (**8**)

Compound **7** (2.0 mmol) in MeOH (10 mL) was added 3 N HCl (2.5 mL) and then heated to 60 °C for 6 h. The mixture was poured in cold NaHCO_3_ (aq) and extracted with EtOAc (3 × 20 mL). The combined organic layers were washed with brine, dried over Na_2_SO_4_, and concentrated under reduced pressure. The crude product was purified by column chromatography on silica (petroleum ether/ethyl acetate 20:1) in order to afford a compound **8** (250 mg, 53%) as a yellow solid. ^1^H-NMR (400 MHz, *d*_6_-DMSO) 13.55 (s, 1H), 10.83 (s, 1H), 6.47 (d, *J* = 10.0 Hz, 1H), 5.90 (s, 1H), 5.52 (d, *J* = 10.0 Hz, 1H), 2.58 (s, 3 H), 1.45 (s, 6H). ^13^C-NMR (100 MHz, *d*_6_-DMSO) 203.0, 165.3, 160.4, 157.0, 125.2, 116.7, 105.1, 102.3, 95.8, 78.3, 33.3, 27.8.

#### 3.1.4. 1,1’-(Methylenebis(5,7-dihydroxy-2,2-dimethyl-2*H*-chromene-6,8-diyl))bis (ethan-1-one) (**9**)

Compound **8** (1.0 mmol) in CH_2_Cl_2_ (10 mL) was added polyformaldehyde (0.5 mmol) and refluxed for 8 h. Solvent was removed under reduced pressure and the residue was purified by column chromatography on silica (petroleum ether/ethyl acetate 200:1) to afford a compound **9** (170 mg, 71%) as a yellow solid. ^1^H-NMR (400 MHz, CDCl_3_) 15.81 (s, 2H), 9.43(s, 2H), 6.63 (d, *J* = 10.0 Hz, 2H), 5.43 (d, *J* = 10.0 Hz, 2H), 3.75 (s, 2H), 2.67 (s, 6H), 1.47 (s, 12H). ^13^C-NMR (100 MHz, CDCl_3_) 202.8, 160.2, 157.3, 154.9, 124.0, 116.0, 104.9, 104.0, 102.4, 77.1, 31.7, 26.9, 14.4. HRMS (ESI) *m*/*z*: calcd for C27H28O8 [M + H]^+^, 481.2656; found, 481.2604.

#### 3.1.5. General Procedure for the Synthesis of Compounds **3** and **4a**–**4h**

Compound **9** (0.5 mmol) in EtOH (3 mL) was added 30% KOH (aq, 1 mL) and substituted benzaldehyde (1.2 mmol) at 0 °C. The reaction was heated to 50 °C for 6 h. After cooling down to 0 °C, 2N HCl was added to the reaction mixture until the pH = 5–6. The precipitate was filtered and then purified by column chromatography on silica to give compounds **3** and **4a**–**4d**.

#### 3.1.6. (2Z,2’Z)-1,1’-(Methylenebis(5,7-dihydroxy-2,2-dimethyl-2*H*-chromene-6,8-diyl))bis (3-phenylprop-2-en-1-one) (**3**)

A red solid (0.266 g, yield 81%). ^1^H-NMR (400 MHz, CDCl_3_) 9.67 (s, 2H), 8.19 (d, *J* = 15.6 Hz, 2H), 7.83 (d, *J* = 15.6 Hz, 2H), 7.62–7.60 (m, 4H), 7.42–7.40 (m, 6H), 6.68 (d, *J* = 12.0 Hz, 2H), 5.47 (d, *J* = 12.0 Hz, 2H), 3.82 (s, 2H), 1.54 (s, 12H). ^13^C-NMR (100 MHz, CDCl_3_) 192.9, 162.8, 158.9, 155.4, 143.3, 135.5, 130.4, 129.0, 128.4, 126.8, 125.1, 117.3, 106.5, 105.3, 103.8, 78.2, 28.0, 15.7.

#### 3.1.7. (2Z,2’Z)-1,1’-(Methylenebis(5,7-dihydroxy-2,2-dimethyl-2*H*-chromene-6,8-diyl))bis (3-(4-methoxyphenyl)prop-2-en-1-one) (**4a**)

A red solid (0.270 g, yield 75%). ^1^H-NMR (400 MHz, CDCl_3_) 9.70 (s, 2H), 8.10 (d, *J* = 15.6 Hz, 2H), 7.83 (d, *J* = 15.6 Hz, 2H), 7.57 (d, *J* = 8.8 Hz, 4H), 6.95 (d, *J* = 8.8 Hz, 4H), 6.69 (d, *J* = 9.6 Hz, 2H), 5.48 (d, *J* = 10.0 Hz, 2H), 3.86 (s, 6H), 3.82 (s, 2H), 1.54 (s, 12H). ^13^C-NMR (100 MHz, CDCl_3_) 192.7, 162.9, 161.6, 158.7, 155.3, 143.4, 130.2, 128.3, 124.9, 124.4, 117.4, 114.5, 106.6, 105.3, 78.0, 55.4, 28.0, 15.8. HRMS (ESI) *m*/*z*: calcd for C43H40O10 [M + H]^+^, 717.3394; found, 717.3366.

#### 3.1.8. (2Z,2’Z)-1,1’-(Methylenebis(5,7-dihydroxy-2,2-dimethyl-2*H*-chromene-6,8-diyl))bis (3-(4-fluoro phenyl)prop-2-en-1-one) (**4b**)

A red solid (0.270g, yield 78%). ^1^H-NMR (400 MHz, CDCl_3_) 9.66 (s, 2H), 8.10 (d, *J* = 15.6 Hz, 2H), 7.79 (d, *J* = 15.6 Hz, 2H), 7.57–7.61 (m, 4H), 7.09-7.13 (d, 4H), 6.69 (d, *J* = 9.6 Hz, 2H), 5.48 (d, *J* = 10.0 Hz, 2H), 3.86 (s, 6H), 3.82 (s, 2H), 1.54 (s, 12H). ^13^C-NMR (100 MHz, CDCl_3_) 192.7, 162.8, 158.9, 155.4, 142.0, 131.8, 130.2, 130.1, 126.6, 125.1, 117.3, 116.3, 116.1, 106.5, 105.2, 103.8, 78.2, 28.0, 15.7. HRMS (ESI) *m*/*z*: calcd for C41H34F2O8 [M + H]^+^, 693.2394; found, 693.2301.

#### 3.1.9. (2Z,2’Z)-1,1’-(Methylenebis(5,7-dihydroxy-2,2-dimethyl-2*H*-chromene-6,8-diyl))bis (3-(furan-3-yl)prop-2-en-1-one) (**4c**)

A white solid (0.253 g, yield 75%). ^1^H-NMR (400 MHz, CDCl_3_) 9.62 (s, 2H), 8.38 (d, *J* = 15.6 Hz, 2H), 7.69 (d, *J* = 15.2 Hz, 2H), 7.61 (d, *J* = 15.2 Hz, 2H), 7.47 (d, *J* = 8.2 Hz, 2H), 7.37 (t, *J* =7.6 Hz, 2H), 7.25(d, *J* = 14.4 Hz, 2H), 7.01 (s, 2H), 6.68 (d, *J* =7.2 Hz, 2H), 5.51 (d, *J* = 10.0 Hz, 2H), 3.83 (s, 2H), 1.60 (s, 12H). ^13^C-NMR (100 MHz, CDCl_3_) δ 192.0, 162.8, 159.0, 155.6, 155.6, 153.7, 129.4, 128.6, 127.5, 126.5, 125.3, 123.3, 121.8, 117.1, 111.8, 111.2, 106.4, 105.4, 103.9, 78.4, 27.8, 15.7. HRMS (ESI) *m*/*z*: calcd for C41H36O8 [M + H]^+^, 657.0782; found, 657.0760.

#### 3.1.10. (2Z,2’Z)-1,1’-(Methylenebis(5,7-dihydroxy-2,2-dimethyl-2*H*-chromene-6,8-diyl))bis (3-(thiophen-3-yl)prop-2-en-1-one) (**4d**)

A red solid (0.271 g, yield 85%). ^1^H-NMR (400 MHz, CDCl_3_) 9.68 (s, 2H), 8.02 (q, *J* = 18.0 Hz, 4H), 7.47–7.29 (m, 2H), 7.09 (d, *J* = 3.6 Hz, 2H), 6.67 (d, *J* = 9.6 Hz, 2H), 5.48 (d, *J* = 9.6 Hz, 2H), 3.81 (s, 2H), 1.56 (s, 12H). ^13^C-NMR (100 MHz, CDCl_3_) 192.0, 162.9, 158.8, 155.3, 141.4, 136.0, 131.9, 128.6, 128.4, 125.9, 125.1, 117.1, 106.5, 105.1, 103.7, 78.2, 27.9, 15.7. HRMS (ESI) *m*/*z*: calcd for C37H32O8S2 [M + H]^+^, 669.3011; found, 669.3024.

#### 3.1.11. (2Z,2’Z)-1,1’-(Methylenebis(5,7-dihydroxy-2,2-dimethyl-2*H*-chromene-6,8-diyl))bis (3-(4-(methoxymethoxy)phenyl)prop-2-en-1-one) (**4e**)

A red solid (0.343g, 88%). ^1^H-NMR (400 MHz, CDCl_3_) 9.69 (s, 2H), 8.11 (d, *J* = 15.6 Hz, 2H), 7.82 (d, *J* = 15.6 Hz, 2H), 7.56 (d, *J* = 8.4 Hz, 4H), 7.08 (d, *J* = 8.8 Hz, 4H), 6.68 (d, *J* = 10.0 Hz, 2H), 5.48 (d, *J* = 10.0 Hz, 2H), 5.22 (s, 4H), 3.82 (s, 2H), 3.49 (s, 6H), 1.53 (s, 12H). ^13^C-NMR (100 MHz, CDCl_3_) 192.7, 162.9, 159.1, 158.7, 155.3, 143.2, 130.1, 129.3, 125.0, 124.9, 117.4, 116.6, 106.6, 105.3, 103.8, 94.2, 78.1, 56.2, 28.0, 15.8.

#### 3.1.12. (2Z,2’Z)-1,1’-(Methylenebis(5,7-dihydroxy-2,2-dimethyl-2*H*-chromene-6,8-diyl))bis (3-(2,4-bis(methoxymethoxy)phenyl)prop-2-en-1-one) (**4f**)

A red solid (0.250g, 56%). ^1^H-NMR (400 MHz, CDCl_3_) 9.71 (s, 2H), 8.26 (d, *J* = 15.6 Hz, 2H), 8.09 (d, *J* = 16.0 Hz, 2H), 7.60 (d, *J* = 8.8 Hz, 2H), 6.87 (d, *J* = 2.4 Hz, 2H), 6.76 (dd, *J* = 8.8, 2.4 Hz, 2H), 6.68 (d, *J* = 10.0Hz, 2H), 5.47 (d, *J* = 10.0 Hz, 2H), 5.27 (s, 4H), 5.20 (s, 4H), 3.82 (s, 2H), 3.52 (s, 6H), 3.50 (s, 6H), 1.52 (s, 12H). ^13^C-NMR (100 MHz, CDCl_3_) 192.9, 163.0, 160.5, 159.0, 158.6, 157.8, 155.3, 138.2, 128.4, 125.0, 124.7, 119.3, 117.4, 109.6, 106.6, 105.3, 103.7, 103.6, 94.8, 94.3, 78.0, 56.4, 56.3, 28.0, 15.8.

#### 3.1.13. (2Z,2’Z)-1,1’-(Methylenebis(5,7-dihydroxy-2,2-dimethyl-2*H*-chromene-6,8-diyl))bis (3-(2,3-bis(methoxymethoxy)phenyl)prop-2-en-1-one) (**4g**)

A red solid (0.367g, 82%). ^1^H-NMR (400 MHz, CDCl_3_) 9.70 (s, 2H), 8.30 (d, *J* = 15.6 Hz, 2H), 8.17 (d, *J* = 16.0 Hz, 2H), 7.34 (dd, *J* = 8.0, 1.2 Hz, 2H), 7.20 (dd, *J* = 8.0, 1.2 Hz, 2H), 7.09 (t, *J* = 8.0 Hz, 2H), 6.67 (d, *J* = 10.0 Hz, 2H), 5.48 (d, *J* = 10.0 Hz, 2H), 5.22 (s, 4H), 5.20 (s, 4H), 3.82 (s, 2H), 3.66 (s, 6H), 3.52 (s, 6H), 1.52 (s, 12H). ^13^C-NMR (100 MHz, CDCl_3_) 192.9, 163.0, 158.8, 155.4, 150.5, 146.5, 138.4, 130.7, 127.7, 125.1, 124.6, 120.1, 118.2, 117.2, 106.6, 105.3, 103.8, 99.5, 95.2, 78.2, 58.0, 56.3, 28.0, 15.8.

#### 3.1.14. (2Z,2’Z)-1,1’-(Methylenebis(5,7-dihydroxy-2,2-dimethyl-2*H*-chromene-6,8-diyl))bis (3-(2,3,4-tris(methoxymethoxy)phenyl)prop-2-en-1-one) (**4h**)

A red solid (0.397g, 76%). ^1^H-NMR (400 MHz, CDCl_3_) 9.72 (s, 2H), 8.24 (d, *J* = 15.6 Hz, 2H), 8.13 (d, *J* = 15.6 Hz, 2H), 7.40 (d, *J* = 9.2 Hz, 2H), 7.01 (d, *J* = 8.8 Hz, 2H), 6.66 (d, *J* = 10.0 Hz, 2H), 5.47 (d, *J* = 10.0 Hz, 2H), 5.25 (s, 4H), 5.21 (s, 4H), 5.16 (s, 4H), 3.81 (s, 2H), 3.65 (s, 6H), 3.62 (s, 6H), 3.52 (s, 6H), 1.52 (s, 12H). ^13^C-NMR (100 MHz, CDCl_3_) 192.7, 163.1, 158.7, 155.3, 153.2, 151.1, 139.7, 138.5, 125.9, 125.0, 124.6, 122.6, 117.3, 112.0, 106.6, 105.2, 103.7, 100.0,98.8, 95.0, 78.1, 58.2, 57.4, 56.4, 28.0, 15.8.

#### 3.1.15. General Procedure for the Synthesis of Compounds **5a**–**5d**

Compounds **4e**–**4h** (0.10 mmol) in MeOH (3 mL) and THF (3 mL) was added 2 N HCl (1 mL) at 0 °C. After addition, the mixture was heated to 45 °C for 24 h. The mixture was poured in cold NaHCO_3_ (aq, 10 mL) and then extracted with EtOAc (3 × 20 mL). The combined organic layers were washed with brine, dried over Na_2_SO_4_, and then concentrated under reduced pressure. The crude product was purified by column chromatography on silica (petroleum ether/ethyl acetate 4:1) to afford a compounds **5a**–**5d**.

#### 3.1.16. (2Z,2’Z)-1,1’-(Methylenebis(5,7-dihydroxy-2,2-dimethyl-2*H*-chromene-6,8-diyl))bis (3-(4-hydroxyphenyl)prop-2-en-1-one) (**5a**)

A red solid (55 mg, 81%). ^1^H-NMR (400 MHz, *d*_6_-DMSO) 15.36 (s, 2H), 10.17 (s, 2H), 7.92 (d, *J* = 15.6 Hz, 2H), 7.72 (d, *J* = 15.6 Hz, 2H), 7.56 (d, *J* = 8.8 Hz, 4H), 6.87 (d, *J* = 8.8 Hz, 4H), 6.63 (d, *J* = 10.0 Hz, 2H), 5.60 (d, *J* = 10.0 Hz, 2H), 3.78 (s, 2H), 1.50 (s, 12H). ^13^C-NMR (100 MHz, *d*_6_-DMSO) 192.5, 163.6, 160.7, 154.5, 143.9, 130.9, 126.4, 125.7, 123.6, 117.2, 116.6, 107.3, 105.2, 103.0, 78.0, 27.9, 16.3. HRMS (ESI) *m*/*z*: calcd for C41H36O10 [M + H]^+^, 689.2181; found, 689.2145.

#### 3.1.17. (2Z,2’Z)-1,1’-(Methylenebis(5,7-dihydroxy-2,2-dimethyl-2*H*-chromene-6,8-diyl))bis (3-(2,4-dihydroxyphenyl)prop-2-en-1-one) (**5b**)

A red solid (50 mg, 70%). ^1^H-NMR (400 MHz, *d*_6_-DMSO) 16.20 (s, 2H), 10.41 (s, 2H), 10.06 (s, 2H), 8.16 (d, *J* = 15.6 Hz, 2H), 7.94 (d, *J* = 15.6 Hz, 2H), 7.36 (d, *J* = 8.8 Hz, 2H), 6.60 (d, *J* = 10.0 Hz, 2H), 6.43 (d, *J* = 2.4 Hz, 2H), 6.34 (dd, *J* = 8.4, 2.0 Hz, 2H), 5.57 (d, *J* = 9.6 Hz, 2H), 3.75 (s, 2H), 1.49 (s, 12H). ^13^C-NMR (100 MHz, *d*_6_-DMSO) 192.8, 163.8, 162.0, 160.2, 154.7, 141.7, 132.52 125.7, 122.7, 117.2, 114.3, 108.7, 107.2, 105.1, 103.2, 103.1, 78.1, 27.7, 16.0. HRMS (ESI) *m*/*z*: calcd for C41H36O12 [M + H]^+^. 721.2279; found, 721.2280.

#### 3.1.18. (2Z,2’Z)-1,1’-(Methylenebis(5,7-dihydroxy-2,2-dimethyl-2*H*-chromene-6,8-diyl))bis (3-(2,3-dihydroxyphenyl)prop-2-en-1-one) (**5c**)

A red solid (52 mg, 73%). ^1^H-NMR (400 MHz, *d*_6_-DMSO) 15.30 (s, 2H), 9.71 (s, 2H), 9.22 (s, 2H), 8.15 (d, *J* = 15.6 Hz, 2H), 7.99 (d, *J* = 15.6 Hz, 2H), 7.02 (d, *J* = 7.2 Hz, 2H), 6.86 (d, *J* = 7.6 Hz, 2H), 6.72 (t, *J* = 7.8 Hz, 2H), 6.61 (d, *J* =10.0 Hz, 2H), 5.55 (d, *J* = 9.6 Hz, 2H), 3.76 (s, 2H), 1.48 (s, 12H). ^13^C-NMR (100 MHz, *d*_6_-DMSO) 193.0, 163.7, 154.6, 146.6, 146.3, 139.8, 126.5, 125.9, 122.7, 119.8, 119.4, 117.6, 117.2, 107.2, 105.3, 103.0, 79.6, 78.2, 27.8, 16.3. HRMS (ESI) *m*/*z*: calcd for C41H36O12 [M − H]^−^, 719.2134; found, 719.2112.

#### 3.1.19. (2Z,2’Z)-1,1’-(Methylenebis(5,7-dihydroxy-2,2-dimethyl-2*H*-chromene-6,8-diyl))bis (3-(2,3,4-trihydroxyphenyl)prop-2-en-1-one) (**5d**)

A red solid (59 mg, 78%). ^1^H-NMR (400 MHz, *d*_6_-DMSO) δ 16.07 (s, 2H), 9.89 (s, 2H), 9.32 (s, 2H), 8.66 (s, 2H), 8.09 (d, *J* = 15.6 Hz, 2H), 7.98 (d, *J* = 15.6 Hz, 2H), 6.93 (d, *J* = 8.8 Hz, 2H), 6.60 (d, *J* = 10.0 Hz, 2H), 6.42 (d, *J* = 8.8 Hz, 2H), 5.56 (d, *J* = 10.0 Hz, 2H), 3.74 (s, 2H), 1.48 (s, 12H). ^13^C-NMR (100 MHz, *d*_6_-DMSO) δ 192.8, 163.7, 154.7, 149.9, 148.5, 133.5, 126.0, 122.7, 121.2, 117.1, 115.2, 108.5, 107.1, 105.2, 103.0, 79.6, 78.2, 27.7, 16.2. HRMS (ESI) *m*/*z*: calcd for C41H35O14 [M − H]^−^, 751.2032; found, 751.2030.

#### 3.1.20. General Procedure for the Synthesis of Compounds **10**–**11**

A solution of KOH (20% aq, 20 mL) and ether (20 mL) was added methyl nitrosourea (1 g) at 0 °C and then stirred for 10 min. Compound **9** (1.0 mmol) in CH_2_Cl_2_ (10 mL) was dripped above ether solution at 0 °C. The reaction mixture was stirred for 6 h at r.t. The reaction was quenched with glacial acetic acid (0.1 mL) and then extracted with EtOAc (3 × 100 mL). The combined organic layers were washed with brine, dried over Na_2_SO_4_, and then concentrated under reduced pressure. The crude product was purified by column chromatography on silica (petroleum ether/ethyl acetate 100:1 and 50:1) to afford compounds **10** and **11**.

#### 3.1.21. 1,1’-(Methylenebis(7-hydroxy-5-methoxy-2,2-dimethyl-2*H*-chromene-6,8-diyl))bis (ethan-1-one) (**10**)

A yellow solid (0.150 g, yield 30%). ^1^H-NMR (400 MHz, CDCl_3_) 13.79 (s, 2H), 6.46 (d, *J* = 10.0 Hz, 2H), 5.46 (d, *J* = 9.6 Hz, 2H), 3.87 (s, 2H), 3.64 (s, 6H), 2.67 (s, 6H), 1.48 (s, 12H).^13^C-NMR (100 MHz, CDCl_3_) 204.1, 163.7, 161.2, 155.0, 125.4, 117.7, 114.4, 108.0, 106.2, 65.8, 61.8, 33.5, 27.7, 15.2.

#### 3.1.22. 1-(6-((8-Acetyl-5,7-dimethoxy-2,2-dimethyl-2H-chromen-6-yl)methyl)-7-hydroxy-5-methoxy-2,2-dimethyl-2*H*-chromen-8-yl)ethan-1-one (**11**)

A yellow solid (0.230 g, yield 44%). ^1^H-NMR (400 MHz, CDCl_3_) 13.83 (s, 1H), 6.49 (d, *J* = 10.0 Hz, 1H), 6.44 (d, *J* = 10.0 Hz, 1H), 5.57-5.53 (d, *J*= 10.0 Hz, 1H), 5.47 (d, *J* = 10.0 Hz, 1H), 3.91 (s, 2H), 3.61 (s, 6H), 3.52 (s, 3H), 2.67 (d, *J* = 5.6 Hz, 3H), 2.50 (s, 3H), 1.48 (s, 6H), 1.40 (s, 6H). ^13^C-NMR (100 MHz, CDCl_3_) 204.1, 201.9, 163.6, 161.1, 156.4, 156.3, 155.1, 149.1, 129.0, 125.5, 121.4, 120.6, 117.6, 117.1, 114.5, 111.2, 107.9, 106.3, 77.5, 76.4, 62.9, 62.1, 61.6, 33.5, 32.6, 27.6, 27.6, 17.6.

#### 3.1.23. (2Z,2’Z)-1,1’-(Methylenebis(7-hydroxy-5-methoxy-2,2-dimethyl-2*H*-chromene-6,8-diyl))bis(3-phenylprop-2-en-1-one) (**12**)

Compound **10** (0.2 mmol) in EtOH (3 mL) was added 30% KOH (aq, 1 mL) and benzaldehyde (0.48 mmol). The reaction mixture was stirred at 50 °C for 6 h. After cooling down to 0 °C, 2 N HCl was added to the reaction mixture until the pH 5–6. The precipitate was filtered and dried to give compound **12** (104 mg, 88%) as a red solid. ^1^H-NMR (400 MHz, CDCl_3_) 13.95 (s, 2H), 8.07 (d, *J* = 15.6Hz, 2H), 7.76 (d, *J* =15.6 Hz, 2H), 7.62–7.57 (m, 4H), 7.46-7.34 (m, 6H), 6.48 (t, *J* = 11.4 Hz, 2H), 5.53-5.46 (m, 2H), 3.98 (s, 2H), 3.68 (s, 6H), 1.54 (s, 12H). ^13^C-NMR (100 MHz, CDCl_3_) 193.7, 164.3, 161.4, 154.4, 142.1, 135.6, 130.0, 128.9, 128.2, 127.9, 125.4, 117.8, 114.8, 108.5, 106.5, 61.9, 27.8, 17.0. HRMS (ESI) *m*/*z*: calcd for C_43_H_40_O_8_ [M + H]^+^, 685.3995; found, 685.3970.

#### 3.1.24. (Z)-1-(6-((5,7-Dimethoxy-2,2-dimethyl-8-((Z)-3-phenylacryloyl)-2*H*-chromen-6-yl)methyl)-7-hydroxy-5-methoxy-2,2-dimethyl-2*H*-chromen-8-yl)-3-phenylprop-2-en-1-one (**13**)

Compound **11** (0.19 mmol) in EtOH (3 mL) was added 30% KOH (aq, 1 mL) and benzaldehyde (0.46 mmol). The reaction mixture was stirred at 50 °C for 6 h. After cooling down to 0 °C, 2 N HCl was added to the reaction mixture until the pH 5-6. The precipitate was filtered and dried to give compound **13** (117 mg, 87%) as a red solid. ^1^H-NMR (400 MHz, CDCl_3_) 14.00 (s, 1H), 8.07 (d, *J* = 15.6 Hz, 1H), 7.77 (d, *J* = 15.6 Hz, 1H), 7.60 (dd, *J* = 5.6, 1.8 Hz, 2H), 7.52 (dd, *J* = 6.6, 3.6 Hz, 2H), 7.44–7.39 (m, 4H), 7.39–7.32 (m, 5H), 7.03–6.97 (m, 1H), 6.55–6.50 (m, 1H), 6.49 (d, *J* = 9.8 Hz, 1H), 5.57-5.53 (m, 1H), 5.53–5.49 (m, 1H), 3.98 (s, 2H), 3.66 (s, 3H), 3.62 (s, 3H), 3.60 (s, 3H), 1.53 (s, 6H), 1.34 (s, 6H). ^13^C-NMR (100 MHz, CDCl_3_) 194.4, 193.7, 164.3, 161.3, 157.1, 156.5, 154.5, 149.8, 144.9, 142.2, 135.5, 134.9, 130.2, 130.1, 129.0, 128.9, 128.8, 128.8, 128.4, 128.2, 127.7, 125.6, 120.4, 119.1, 117.8, 117.2, 115.0, 111.2, 108.5, 106.6, 77.5, 76.3, 62.6, 62.1, 61.7, 27.8, 27.6, 22.6, 17.8, 14.1. HRMS (ESI) *m*/*z*: calcd for C44H42O8 [M + Na]^+^, 733.2785; found, 733.2770.

### 3.2. α-Glucosidase Inhibition Assay

The commercially available α-glucosidase from baker’s yeast (Sigma, G5003) was selected as the target protein while using *p*-nitrophenyl-α-d-glucopyranoside (*p*NGP, Sigma, N1377) as the substrate. The compounds and acarbose were dissolved in DMSO. The enzyme and substrate were dissolved in potassium phosphate buffer (0.05 M, pH 6.8). The enzymatic reaction mixture that was composed of α-glucosidase (0.04 U, 20 μL), substrate (0.5 mM, 30 μL), test compounds (20 μL), and potassium phosphate buffer (130 μL) was incubated at 37 °C for 30 min. The enzymatic activity was detected by spectrophotometer at the wavelength of 405 nm. The results are the average of three independent experiments, each being performed in duplicate [8].

### 3.3. DPP-4 Inhibition Assay

The commercially available DPP-4 (PROSPEC, ENZ-375) was selected as the target protein using glycyl proline *p*-nitroaniline as the substrate. The compounds and sitagliptin were dissolved in DMSO. The enzyme and substrate were dissolved in Tris-HCl buffer (0.1 M, pH 8.0). The enzymatic reaction mixture that was composed of DPP-4 (0.01 U, 50 μL), substrate (2.2 mM, 50 μL), test compounds (10 μL), and Tris-HCl buffer (90 μL) was incubated at 37 °C for 30 min. The enzymatic activity was detected by spectrophotometer at the wavelength of 405 nm. The results are the average of three independent experiments, each being performed in duplicate [12].

### 3.4. Kinetics of Enzyme Inhibition

The kinetics of enzyme inhibition was performed with the same assays detailed above, in the presence of different concentrations of compounds **4d** and **5d**, substrate, and enzyme, respectively. The inhibitory kinetics of **4d** and **5d** on α-glucosidase was analyzed using the Lineweaver-Burk plot of the substrate concentration and velocity.

### 3.5. Glucose Consumption Assay

Briefly, the HepG2 cells were cultured in Dulbecco’s-modified Eagle’s medium (DMEM) containing 25 mmol/L d-glucose, 10% heat-inactivated fetal bovine serum (FBS), 10 U/mL penicillin, and 10 mg/mL streptomycin at 37 °C, 5% CO_2_ atmosphere. The culture solution was replaced every other day and then passaged once for 2–3 d. The cells were seeded into 96-well plate with twenty-four wells left as blanks. After reaching 70–80% confluence, the cells were washed with PBS twice and the medium was replaced by DMEM containing 25 mmol/L d-glucose to hunger for 24 h. The cells were washed with PBS twice and the medium was replaced by RPMI-1640 containing 11.1 mmol/L glucose that was supplemented with 0.2% bovine serum albumin (BSA). To the medium was then added 1 mmol/L metformin or purified product at different concentrations, and DMSO was used as the blank control. The glucose concentration in the medium determined a DMSO after 24 h treatment. Glucose consumption = glucose concentrations of blank wells − glucose concentrations of plated wells. The sulforhodamine B (SRB) assay was used to adjust the glucose consumption values. The SRB assay is based on the measurement of cellular protein content. After an incubation period, the cells were fixed with 10%, 4 °C trichloroacetic acid, stained for 1 h at 4 °C, and then washed with purified water. After drying, 60 μL SRB (3 mg/mL) was added into each well and then left at room temperature for 30 min. SRB was dissolved by 1% acetic acid before air drying. Bound SRB was solubilized with 150 μL Tris-base (10 mmol/L, pH 10.5) solution for OD determination at 546 nm while using a microplate reader [13].

## 4. Conclusions

A series of rottlerone analogues were synthesized and evaluated for their α-glucosidase and DPP-4 inhibitory activities. Compounds **4d** and **5d** were identified as new potent α-glucosidase inhibitors and moderate DPP-4 inhibitors. The kinetic analysis showed that compounds **4d** and **5d** are irreversible α-glucosidase inhibitors, and that **5d** acts in a mixed competitive inhibitory mode. In addition, compounds **4d** and **5d** showed glucose consumption-promoting activity in HepG2 cells. Of these two compounds, the activity of **5d** at 0.2 μM was comparable to that of metformin at a concentration of 1 mM. In view of these significant results, compound **5d**, which combines both α-glucosidase and DPP-4 inhibitory activities, presents great potential as a lead compound for the development of a novel dual inhibitor treatment strategy for type 2 diabetes.

## Data Availability

There are no data supporting.

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
