# Peer review of "Synthesis of Rottlerone Analogues and Evaluation of Their α-Glucosidase and DPP-4 Dual Inhibitory and Glucose Consumption-Promoting Activity"

_molecules, 2021, doi:10.3390/molecules26041024_

Round 1
Reviewer 1 Report
The paper by Zhang et al. is focused on the synthesis of rottlerone analogues and the assessment of their α-glucosidase and DPP-4 dual inhibitory and glucose consumption- promoting activity. Among the studied compounds two best candidates (4d and 5d) were identified as new potent α-glucosidase inhibitors and moderate DPP-4 inhibitors. In my opinion this manuscript is well written, the study design is thorough, and the results are promising. The publication of this paper will be in the great interest of medicinal chemistry community especially for those involved in the development of novel potential anti-diabetic agents.
Listed below are some suggested changes:
- Figure 1: please complete the name of compound 2 and try to specify R and R’ within structure denoted as “this work“
- Line 71: “scheme 1“ should be “Scheme 1”
- Lines 106-108: this sentence should be improved to become more clear
- Line 125: “Figure 2, a and b” should be “Figure 2a and b”
- References should be carefully checked and unified (e.g. position 12 – lack of the names of the authors; full journals names vs abbrevs)
Reviewer 2 Report
This is a well written account of some novel glucosidase inhibitors. Both the chemistry and biological evaluation is properly described and is worthy of publication in its present form.
In the nomenclature title of compounds with the stereochemical descriptors Z,Z, an uppercase letter is require for the first substantive word - for example in 3.1.6 ...(2Z,2'Z)-1,1'-(methylen...., should be (2Z,2'Z)-1,1'-(Methylen....and in all the subsequent compounds
All final products had a purity of ≥ 152 95%.
Author Response
Point 1: In the nomenclature title of compounds with the stereochemical descriptors Z,Z, an uppercase letter is require for the first substantive word - for example in 3.1.6 ...(2Z,2'Z)-1,1'-(methylen...., should be (2Z,2'Z)-1,1'-(Methylen....and in all the subsequent compounds
Response 1: All uppercase letters have been revised for the first substantive words from 3.1.6 to 3.1.24.
Point 2: All final products had a purity of ≥ 152 95%.
Response 2: In line 152, "All final products had a purity of ≥ 95%. " has been revised as "Compound purity was determined by UV absorbance at 254 nm during tandem liquid chromatography/mass spectrometry (Agilent, Santa Clara, CA, USA). All final products had a purity of ≥ 95% as determined using this method. "
Reviewer 3 Report
After reading the manuscript titled, "Synthesis of rottlerone analogues and evaluation of their α-glucosidase and DPP-4 dual inhibitory and glucose consumption-promoting activity", by Zhang et al., I do not find any major issues that require attention in the manuscript. Overall, the manuscript is well described and although is a preliminary investigation report, I recommend to be accepted for publication.
Author Response
Thank you very much.